# Amortized-Precision Quantization for Early Exiting in Vision Transformers

## Abstract

Vision Transformers (ViTs) achieve state-of-the-art results across classification, detection, and segmentation, but their heavy computation hinders deployment on resource-constrained devices. Quantization is a common technique to improve efficiency, yet conventional approaches assume static inference and ignore the input-dependent utility of layers under dynamic strategies such as Early Exiting (EE). This mismatch leads to inefficient bit allocation: shallow layers may be over-provisioned while deeper exits, which dominate late-stage decisions, remain under-optimized. We introduce **Amortized-Precision Quantization (APQ)**, a new perspective that treats precision as a utilization-dependent resource, exposing depth–precision and shallow-deep trade-offs. Building on APQ, we propose **Mutual Adaptive Quantization with Early Exiting (MAQEE)**, a bi-level optimization framework that jointly calibrates exit thresholds and reallocates bit-widths under risk control. We theoretically establish MAQEE's superiority over static quantization in dynamic inference, and empirically show that it reduces bit operations by up to 95% while preserving accuracy, outperforming strong baselines by as much as 20% on ViT classification, detection, and segmentation benchmarks.

## 1 Introduction

Vision Transformers (ViTs) (Dosovitskiy et al., 2020) have achieved remarkable success in image classification (Chen et al., 2021), object detection (Carion et al., 2020), and semantic segmentation (Gao et al., 2021). Despite these advances, their large parameter counts and high computational demands hinder deployment on edge devices such as smartphones and IoT systems (Zheng et al., 2023; Shang et al., 2024). To improve efficiency in resource-constrained environments, quantization (Courbariaux et al., 2015) has become a widely adopted technique, reducing memory footprint and inference latency by lowering the bit-width of weights and activations. Early efforts typically adopt *Fixed-Precision Quantization (FPQ)* (Krishnamoorthi, 2018; Jacob et al., 2018; Yang et al., 2019), which assigns a uniform bit-width to all layers. However, because ViT layers usually differ in weight distributions (Liu et al., 2021a) and robustness to quantization noise (Tai et al., 2024), FPQ often yields suboptimal accuracy–efficiency trade-offs (Gholami et al., 2022). To overcome this, *Mixed-Precision Quantization (MPQ)* (Xiao et al., 2023; Jeon et al., 2024) assigns bit-widths in a fine-grained manner by adapting to the quantization sensitivity of each layer, allowing more efficient hardware utilization under tight resource budgets.

Although MPQ mitigates the limitations of FPQ, it still assumes a *static inference path*, executing all layers end-to-end. As ViTs scale to support diverse tasks, modern architectures increasingly adopt *dynamic inference strategies* such as Early Exiting (EE) (Teerapittayanon et al., 2016; Laskaridis et al., 2021; Xu et al., 2023) to adapt computation dynamically to input complexity. Specifically, EE attaches lightweight exit heads to intermediate layers of ViT backbones, enabling high-confidence samples to terminate early and thereby reduce latency. However, conventional quantization interacts poorly with EE because it calibrates precision using training-time activation statistics, under the assumption of a stable inference path. In contrast, EE introduces input-dependent execution (Rahmath P et al., 2024) that violates this assumption, leading to unstable precision requirements and unreliable confidence estimates. Our preliminary study further substantiates this incompatibility, showing that quantizing EE-trained models reduces accuracy by up to 50%, whereas applying EE to quantized models results in unreliable exits and limited efficiency gains.

Motivated by these observations, we propose **Amortized-Precision Quantization (APQ)**, a quantization paradigm tailored for dynamic inference. Unlike conventional methods that statically assign bit-widths, APQ treats precision as a dynamic resource amortized across layers based on their utilization. We analyze its sources of error from two perspectives. At the global level, a **depth–precision trade-off** arises: allocating fewer bits enables deeper inference under a fixed budget but increases quantization error, while higher precision reduces noise but limits depth and may trigger premature or delayed exits. At the local level, a **shallow–deep trade-off** emerges: allocating more bits to shallow layers reduces early misclassifications but overuses resources, whereas deeper layers are more efficient but incur the risk of error propagation. We theoretically prove that naively combining quantization with early exiting yields suboptimal performance for APQ.

To this end, we introduce **Mutual Adaptive Quantization with Early Exiting (MAQEE)**, a unified bi-level optimization framework for APQ. In the outer loop, MAQEE calibrates input-adaptive exit thresholds by minimizing *Boundary Sensitivity Risk (BSR)*, which penalizes unstable decisions that can be easily flipped under quantization noise. In the inner loop, it reallocates layer-wise bit-widths by balancing early-exit risks against quantization distortions, measured by *Quantization-Induced Drift (QID)*. QID captures global distributional shifts in intermediate representations and reflects the dual requirement of exit heads to support both deeper propagation and premature exits. These two loops are integrated into an iterative routine that alternates threshold calibration, risk-guided precision reallocation, and lightweight self-distillation recovery, thereby progressively stabilizing accuracy under low-bit execution. By tightly coupling quantization with early exiting, MAQEE enables efficient, low-latency inference without sacrificing accuracy.

**Contributions.** Our contributions are fourfold: (1) we formalize the incompatibility between conventional quantization and dynamic inference and provide theoretical proof, introducing *Amortized-Precision Quantization (APQ)* as a new paradigm that aligns bit-width allocation with layer utilization under early exiting; (2) we present *Mutual Adaptive Quantization with Early Exiting (MAQEE)*, a unified bi-level framework that integrates risk-controlled thresholding, utilization-guided bit-width allocation, and progressive self-distillation for efficient APQ; (3) we provide theoretical guarantees that MAQEE outperforms static FPQ/MPQ under early exits; and (4) MAQEE reduces BOPs by 95% while preserving task accuracy, and outperforms strong quantization and early-exiting baselines by at least 20% on multiple ViT backbones across three core vision tasks.

## 2 RELATED WORK

Vision Transformers (ViTs) (Dosovitskiy et al., 2020) effectively capture long-range dependencies through self-attention (Vaswani et al., 2017; Raghu et al., 2021), but their deep architectures and large token sizes result in substantial computational and memory costs, restricting deployment in real-time or resource-constrained scenarios (Zheng et al., 2023; Shang et al., 2024).

A prominent line of research mitigates these costs through quantization, which reduces weights and activations to lower bit-widths (Courbariaux et al., 2015; Krishnamoorthi, 2018). Early efforts relied on *Fixed-Precision Quantization (FPQ)* (Jacob et al., 2018; Yang et al., 2019), applying a uniform bit-width across all layers. However, since ViT layers exhibit heterogeneous weight distributions (Liu et al., 2021a) and varying robustness to quantization noise (Tai et al., 2024), FPQ often fails to achieve an optimal accuracy–efficiency trade-off (Gholami et al., 2022). To overcome this limitation, *Mixed-Precision Quantization (MPQ)* allocates different bit-widths to different layers based on sensitivity (Xiao et al., 2023; Jeon et al., 2024). Recent studies further refine ViT quantization via both Post-Training Quantization (PTQ) (Liu et al., 2021c; Ding et al., 2022; Li et al., 2023; Shi et al., 2024) and Quantization-Aware Training (QAT) (Wang et al., 2025; Nagel et al., 2022; Li et al., 2022), aiming to preserve accuracy under low-bit constraints.Despite these advances, existing quantization approaches mainly assume a *static inference path*, where all layers are executed end-to-end, leaving their compatibility with dynamic inference largely unexplored.

Another complementary direction is *dynamic inference*, which adapts computation to input complexity (Chen et al., 2025; Riquelme et al., 2021; Hwang et al., 2023) by activating only a subset of the network, thereby reducing unnecessary computation, inspired by model pruning (Han et al., 2015). Among such methods, Early Exiting (EE) (Xin et al., 2020; Schuster et al., 2022) attaches lightweight exit heads to intermediate layers, allowing high-confidence samples to terminate early and reduce inference cost. In the context of ViTs, Bakhtiarnia et al. (2021) first incorporated multi-

ple exit branches into the backbone, while Xu et al. (2023) enhanced intermediate feature quality by combining local perception with global attention, achieving a better accuracy–efficiency trade-off. More recently, Rahmath P et al. (2024) proposed adaptive token routing and hierarchical EE architectures, further underscoring the potential of dynamic execution. However, the effectiveness of EE hinges on reliable confidence estimation, which is highly susceptible to quantization noise and often results in premature or delayed exits.

Recently, several studies have explored combining quantization with early exiting. Saxena & Roy (2023) integrated mixed-precision quantization into CNN exits, but its per-layer learned parameters hinder scalability to large ViTs. Regol et al. (2024) enabled per-sample exit paths by maintaining multiple model precisions, at the cost of high memory usage. In contrast, we theoretically analyze the interference between quantization and early exiting, and propose *Amortized-Precision Quantization*, a unified framework that jointly optimizes bit allocation and exit thresholds.

## 3 PROBLEM FORMULATION

Here, we formally introduce Amortized-Precision Quantization and formulate its corresponding optimization problem.

**Quantization.** We first briefly review quantization (Jacob et al., 2018), which replaces floating-point operations with low-bit integer operations, thereby reducing both memory footprint and inference cost. Specifically, quantization uses a scale factor $s$ and zero-point $t$ to map values into $[0, 2^b - 1]$, with dequantization $s(x^q - t)$ used to approximately recover the original value. Here, $b$ denotes the bit-width. Depending on how bit-widths are assigned across layers, two strategies are widely studied: *Fixed-Precision Quantization (FPQ)* (Jacob et al., 2018; Yang et al., 2019), where all layers share the same bit-width, and *Mixed-Precision Quantization (MPQ)* (Xiao et al., 2023; Jeon et al., 2024), where different layers may use different bit-widths.

Let a network consist of $L$ layers, and let $\mathcal{B}$ denote the set of admissible bit-widths (e.g., $\{2, 4, 8\}$). Each layer $\ell$ is assigned a bit-width $b_\ell \in \mathcal{B}$, and we denote the allocation by $\mathbf{b} = \{b_\ell\}_{\ell=1}^L$. The computational cost of layer $\ell$ on input $x$ under bit-width $b_\ell$ is written as $c_\ell(x, b_\ell)$, and the total complexity is given by $\sum_{\ell=1}^L c_\ell(x, b_\ell)$. Following prior works (Shang et al., 2024; Chen et al., 2025), we measure this complexity in terms of *bit operations (BOPs)*, which quantify multiplications and additions under quantized precision. BOPs are defined as FLOPs $\times B_w \times B_a$, where $B_w$ and $B_a$ denote the bit-widths of weights and activations. Since FPQ and MPQ require every input to traverse all $L$ layers, the expected complexity depends solely on the allocation $\mathbf{b}$.

**Early Exiting (EE).** In parallel, early exiting improves efficiency by shortening the inference path through attaching auxiliary classifiers (exit heads) to intermediate layers, allowing samples to terminate computation early once predictions become sufficiently confident (Schuster et al., 2022). This mechanism yields input-dependent depth: easy samples exit early, while harder ones propagate deeper. Formally, at layer $\ell$, let $z_\ell(x)$ denote the logits before softmax from the exit head for input $x$, and define the exit confidence as $\rho_\ell(x) := \max_j \mathrm{softmax}(z_\ell(x))_j$. The input exits at layer $\ell$ if $\rho_\ell(x) \geq \phi_\ell$, where $\phi_\ell \in [0, 1]$ is the confidence threshold for that layer; otherwise, it continues until the final classifier at layer $L$. The objective of EE is to jointly train the exit heads and optimize the thresholds $\boldsymbol{\phi} = \{\phi_\ell\}_{\ell=1}^L$ across layers.

To quantify the effectiveness of early-exiting, we define the sample-wise utilization indicator $u_\ell(x, \boldsymbol{\phi}) = \mathbf{1}\{x \text{ reaches layer } \ell \mid \boldsymbol{\phi}\}$, which indicates whether input $x$ is processed by layer $\ell$ under thresholds $\boldsymbol{\phi}$. Taking the expectation over the data distribution $\mathcal{D}$ gives the utilization factor $u_\ell(\boldsymbol{\phi}) = \mathbb{E}_{x \sim \mathcal{D}}[u_\ell(x, \boldsymbol{\phi})]$, which quantifies the average probability that a layer is executed. The expected inference depth is then defined as $T(\boldsymbol{\phi}) = \sum_{\ell=1}^L u_\ell(\boldsymbol{\phi})$, which characterizes the average number of layers traversed per input. The objective of EE is to learn thresholds $\boldsymbol{\phi}$ that reduce the expected depth $T(\boldsymbol{\phi})$, thereby lowering computation cost while preserving prediction accuracy.

**Amortized-Precision Quantization (APQ).** While conventional quantization schemes assume a static inference path, they overlook the fact that modern architectures increasingly adopt dynamic inference, where execution depth varies across inputs due to mechanisms such as early exiting. This mismatch results in unstable precision requirements under input-dependent execution. To address this issue, we propose *Amortized-Precision Quantization (APQ)*, which allocates precision (bit-

width) according to the *utility* of each layer—namely, its execution frequency—thereby enabling robust and efficient inference.

**Definition 1** (Amortized-Precision Quantization (APQ)). *Consider a network with $L$ layers, where each layer $\ell$ is assigned a bit-width $b_\ell \in \mathcal{B}$, with $\mathcal{B}$ denoting the set of admissible bit-widths. The amortized computational complexity of a bit allocation $\boldsymbol{b}$ under an early-exit policy $\boldsymbol{\phi}$ is defined as*

$$Complexity_{APQ}(\boldsymbol{b}, \boldsymbol{\phi}) = \sum_{\ell=1}^{L} \mathbb{E}_{x \sim \mathcal{D}}[u_\ell(x, \boldsymbol{\phi}) \cdot c_\ell(x, b_\ell)] = \sum_{\ell=1}^{L} u_\ell(\boldsymbol{\phi}) \cdot \mathbb{E}_{x \sim \mathcal{D}}[c_\ell(x, b_\ell)], \quad (1)$$

*where $c_\ell(x, b_\ell)$ denotes the computational cost of layer $\ell$ under bit-width $b_\ell$, and $u_\ell(x, \boldsymbol{\phi}) \in \{0, 1\}$ indicates whether layer $\ell$ is executed for input $x$ given thresholds $\boldsymbol{\phi}$. Taking the expectation over the data distribution $\mathcal{D}$ yields the* utilization factor

$$u_\ell(\boldsymbol{\phi}) = \mathbb{E}_{x \sim \mathcal{D}}[u_\ell(x, \boldsymbol{\phi})]. \quad (2)$$

Since the utilization factor $u_\ell(\boldsymbol{\phi})$ depends on the input $x$, an effective APQ algorithm must jointly optimize the bit-width allocation $\boldsymbol{b}$ and exit thresholds $\boldsymbol{\phi}$ to balance accuracy and efficiency. Unlike MPQ, which may waste precision on rarely used layers, APQ aligns bit allocation with layer utility, concentrating precision where it most affects performance. Formally, this yields

$$(\boldsymbol{b}^*, \boldsymbol{\phi}^*) = \arg\min_{\boldsymbol{b}, \boldsymbol{\phi}} \ \mathbb{E}_{(x,y) \sim \mathcal{D}}\big[\mathcal{L}_{\mathrm{CE}}\big(f_{\theta, \boldsymbol{b}, \boldsymbol{\phi}}(x), y\big)\big] + \lambda \cdot Complexity_{APQ}(\boldsymbol{b}, \boldsymbol{\phi}), \quad (3)$$

where $\mathcal{L}_{\mathrm{CE}}$ denotes the cross-entropy loss, $f_{\theta, \boldsymbol{b}, \boldsymbol{\phi}}$ is the quantized network with parameters $\theta$, bit allocation $\boldsymbol{b}$, and early-exit policy $\boldsymbol{\phi}$, and $\lambda$ balances the trade-off between accuracy and efficiency.[1]

# 4 MUTUAL ADAPTIVE QUANTIZATION WITH EARLY EXITING

To effectively solve APQ, two key allocation trade-offs must be addressed. At the global level, the *depth–precision* trade-off balances inference depth against quantization error. At the local level, the *shallow–deep* trade-off weighs early-exit accuracy against resource efficiency. These coupled trade-offs motivate a bi-level optimization framework, **Mutual Adaptive Quantization with Early Exiting (MAQEE)**, which jointly optimizes exit thresholds and bit-width allocation. A detailed pseudocode is provided in Appendix D.

## 4.1 BI-LEVEL OPTIMIZATION WITH AMORTIZED-PRECISION QUANTIZATION

Since the objective of APQ is to minimize the expected amortized complexity while preserving prediction accuracy, the problem is naturally formulated as a **bi-level optimization** over $(\boldsymbol{b}, \boldsymbol{\phi})$.

Recall that $u_\ell(\boldsymbol{\phi})$ denotes the probability that inference reaches layer $\ell$, and $T(\boldsymbol{\phi}) = \sum_{\ell=1}^{L} u_\ell(\boldsymbol{\phi})$ is the expected number of executed layers.

**Outer-loop (exit-threshold optimization).** Let $\boldsymbol{\phi} = \{\phi_\ell\}_{\ell=1}^{L}$ denote the confidence thresholds of each exit head. The outer problem optimizes these thresholds to balance accuracy and latency:

$$\boldsymbol{\phi}^* = \arg\min_{\boldsymbol{\phi}} \ \mathbb{E}_{(x,y) \sim \mathcal{D}}\big[\mathcal{L}_{\mathrm{CE}}\big(f_{\theta, \boldsymbol{b}^*(\boldsymbol{\phi}), \boldsymbol{\phi}}(x), y\big)\big] + \lambda_{\mathrm{outer}} T(\boldsymbol{\phi}), \quad (4)$$

where $\lambda_{\mathrm{outer}}$ is a regularization parameter that controls the trade-off between accuracy and latency, and $\boldsymbol{b}^*(\boldsymbol{\phi})$ denotes the optimal bit-width allocation given thresholds $\boldsymbol{\phi}$, which in turn determine the depth of the execution path.

**Inner-loop (bit-width allocation).** For fixed exit thresholds $\boldsymbol{\phi}$, the bit-width allocation $\boldsymbol{b} = \{b_\ell\}_{\ell=1}^{L}$ is chosen to minimize amortized complexity while preserving accuracy:

$$\boldsymbol{b}^* = \arg\min_{\boldsymbol{b}} \ \mathbb{E}_{(x,y) \sim \mathcal{D}}\big[\mathcal{L}_{\mathrm{CE}}\big(f_{\theta, \boldsymbol{b}, \boldsymbol{\phi}}(x), y\big)\big] + \lambda_{\mathrm{inner}} Complexity_{APQ}(\boldsymbol{b}, \boldsymbol{\phi}), \quad (5)$$

where $\lambda_{\mathrm{inner}}$ is a regularization parameter that controls the strength of compression. The bi-level optimization explicitly couples quantization with early exiting: (i) layers with high utilization $u_\ell(\boldsymbol{\phi})$

---

[1]We theoretically prove that naively combining quantization with early exiting degrades performance: (i) quantization noise can distort exit signals, leading to premature or delayed exits (Theorem 1); and (ii) early exiting, in turn, amplifies the accumulation of quantization errors across layers (Theorem 2).

are assigned larger bit-widths to preserve fidelity; (ii) layers that are rarely executed are quantized more aggressively to reduce cost; and (iii) exit thresholds $\phi$ are optimized while accounting for quantization-induced noise. By jointly optimizing both $\phi$ and $\boldsymbol{b}$, APQ achieves a balanced trade-off among accuracy, efficiency, and stability in dynamic inference. We prove that APQ is NP-hard (detailed in Theorem 4 in Appendix A).

In the following, we detail the optimization of exit thresholds (outer loop) in Section 4.2 and the allocation of bit-widths (inner loop) in Section 4.3. Section 4.4 then introduces the full iterative optimization procedure, which incorporates a self-distillation method for precision recovery.

## 4.2 Outer Loop: Exit-Threshold Optimization

The outer loop aims to optimize the exit thresholds $\phi$ under a fixed bit allocation $\boldsymbol{b}$. A higher threshold $\phi_\ell$ improves reliability but increases computational cost, whereas a lower threshold reduces cost at the risk of accuracy loss and greater instability under quantization. However, directly estimating $\phi$ via gradient descent is computationally expensive, as this requires jointly training the exit heads and optimizing the thresholds (Rahmath P et al., 2024). Instead, conventional early-exit methods adopt a surrogate risk function to evaluate candidate thresholds. In particular, they optimize the **Performance Gap Risk (PGR)** (Jazbec et al., 2024), defined as

$$\mathrm{PGR}_\ell(\phi) = \mathbb{E}_{(x,y)\sim\mathcal{D}}\big[\mathcal{L}_{\mathrm{CE}}(\hat{y}_\ell, y) - \mathcal{L}_{\mathrm{CE}}(\hat{y}, y)\big], \tag{6}$$

where $\hat{y}_\ell$ is the prediction from the selected exit head, and $\hat{y}$ is the prediction of the full-depth classifier. Here, $\ell$ is determined dynamically by the input $x$ and thresholds $\phi$, i.e., the earliest exit layer whose confidence score $\rho_\ell(x)$ exceeds $\phi_\ell$. Thus, PGR measures the expected excess loss of the chosen early exit relative to the final classifier.[2]

However, PGR only reflects the performance degradation from early exiting and overlooks the instability introduced by quantization, where confidence scores near thresholds are highly sensitive to small perturbations, often leading to premature or delayed exits. To address this, we introduce the **Boundary Sensitivity Risk (BSR)**, which penalizes cases where the confidence score $\rho_\ell(x)$ lies close to the threshold $\phi_\ell$, defined as

$$\mathrm{BSR}_\ell(\phi) = \mathbb{E}_{x\sim\mathcal{D}}\Big[\mathbf{1}\{|\rho_\ell(x) - \phi_\ell| \leq \tau_\ell\}\Big], \tag{7}$$

where $\tau_\ell$ denotes the tolerance margin at layer $\ell$, estimated from the unquantized model by treating early exits as a soft surrogate for the true margin. A high BSR indicates that the exit threshold is placed near a decision boundary, where even small perturbations can flip exit outcomes. Thus, BSR complements PGR by providing a stability-aware signal for threshold optimization.

Accordingly, we rewrite the objective in 4 to obtain the optimal thresholds $\phi$:

$$\phi^\star \in \arg\min_{\phi} \left\{ \frac{1}{L}\sum_{\ell=1}^{L}\Big(\mathrm{PGR}_\ell(\phi) + \mathrm{BSR}_\ell(\phi)\Big) + \lambda_{\mathrm{outer}}\, T(\phi) \right\}, \tag{8}$$

where PGR measures the average excess loss from early exiting, BSR penalizes instability near decision boundaries, and $T(\phi)$ denotes the expected number of executed layers (i.e., amortized inference cost). Finally, MAQEE employs coordinate search to determine the optimal $\phi$. For each input $x$, the confidence score $\rho_\ell(x)$ is obtained with a single feedforward pass, after which candidate thresholds can be efficiently evaluated without iterative retraining or gradient descent (Jazbec et al., 2024). By jointly minimizing performance degradation and boundary sensitivity, the outer loop yields stable confidence thresholds. These optimized thresholds form the basis for the inner loop, which reallocates bit-widths across layers to further reduce amortized complexity.[3]

## 4.3 Inner Loop: Bit-width Allocation

Afterward, the inner loop reallocates bit-widths $\boldsymbol{b}$ conditioned on the exit thresholds $\phi$ determined by the outer loop, with the objective of reducing the expected complexity in APQ. Similar to the exit-threshold optimization in the outer loop, we adopt a risk-based scheme to guide bit allocation

---

[2]For object detection, we adopt a bipartite matching loss in PGR (Fang et al., 2021).

[3]All quantities are estimated on a held-out calibration set (about 5% of the training data), which provides a sufficient approximation of $\mathcal{D}$ for reliable estimates (Li et al., 2022).

in the inner loop. In conventional MPQ (Tai et al., 2024; Li et al., 2023), a standard indicator for layer-wise precision loss is the inverse signal-to-quantization-noise ratio (SQNR), defined as $\text{SQNR}_\ell^{-1} = \frac{||h_\ell||^2}{||h_\ell - \tilde{h}_\ell||^2}$, where $h_\ell$ and $\tilde{h}_\ell$ denote the full-precision and quantized activations (i.e., hidden states) of the $\ell$-th layer for input $x$, respectively. The inverse SQNR quantifies the degradation of signal fidelity under reduced precision: the numerator measures the expected quantization noise power, while the denominator corresponds to the expected signal power.

However, early exiting not only amplifies quantization noise but also alters the activation distribution, since layers must simultaneously support deeper propagation and premature exits, thereby exposing them to greater representation distortion and decision instability. This effect becomes more severe in deeper layers, as quantization errors accumulate across successive transformations, ultimately destabilizing exit predictions. Therefore, instead of focusing solely on bitwise error (i.e., inverse SQNR), we introduce the **Quantization-Induced Drift (QID)**, which measures the discrepancy between the activations of quantized and full-precision models. In general, QID can be defined using a statistical divergence $D(h_\ell, \tilde{h}_\ell)$ between the full-precision activations $h_\ell$ and their quantized counterparts $\tilde{h}_\ell$. However, directly computing divergences such as KL or Wasserstein distance is computationally expensive, as it requires high-dimensional density estimation (Yuan et al., 2024) or solving optimal transport problems (Zhang et al., 2023). To make QID practical, we instantiate it using a lightweight range-ratio approximation that compares the value ranges of quantized and full-precision activations:

$$\text{QID}_\ell = \left( \frac{\max_i \tilde{h}_\ell(x)_i - \min_i \tilde{h}_\ell(x)_i}{\max_i h_\ell(x)_i - \min_i h_\ell(x)_i} - 1 \right)^2, \tag{9}$$

which captures value-range drift. If quantization alters the output distribution, the dynamic range of $\tilde{h}_\ell$ deviates from that of $h_\ell$. Thus, a larger QID reflects stronger quantization-induced drift, which undermines both representation quality and the stability of early-exit decisions, effectively replacing the extreme-value rule with noise-driven behavior during quantization.

Importantly, we do not regularize complexity in 5, as this would bias the allocation toward minimizing cost while neglecting the instability and accuracy loss caused by low-precision operations and premature exits. Following MPQ (Tai et al., 2024), we adopt a fixed bit budget and optimize the allocation $\boldsymbol{b}$ by introducing per-layer risk scores $R_\ell$, which combine decision-level risks from early exiting (PGR and BSR) with representation-level risks from quantization (QID and $\text{SQNR}^{-1}$):

$$R_\ell = \alpha\big(\text{BSR}_\ell + \text{PGR}_\ell\big) + (1 - \alpha)\big(\text{QID}_\ell + \text{SQNR}_\ell^{-1}\big), \tag{10}$$

where $\alpha \in [0, 1]$ controls the trade-off between decision-level and representation-level risks.

To optimize the bit allocation $\boldsymbol{b}$, we normalize $R_\ell$ by the marginal expected BOPs:

$$\Psi_\ell = \frac{R_\ell}{u_\ell(\boldsymbol{\phi}) \cdot \Delta\text{BOPs}_\ell / \Delta b_\ell + \varepsilon}, \tag{11}$$

where $u_\ell(\boldsymbol{\phi}) = \mathbb{E}_{x \sim \mathcal{D}}[u_\ell(x, \boldsymbol{\phi})]$ denotes the *layer utility*, i.e., the expected probability that an input reaches layer $\ell$ under thresholds $\boldsymbol{\phi}$. Here, $\Delta\text{BOPs}_\ell / \Delta b_\ell$ is the marginal increase in bit-operations when the bit-width of layer $\ell$ is raised by one unit, with $\varepsilon$ added for numerical stability.

By normalizing risk with its marginal contribution to the APQ complexity, $\Psi_\ell$ accounts for both utility and bit-wise cost, ensuring that bits are not wasted on low-utility layers while prioritizing those that incur high instability per unit complexity. In practice, MAQEE iteratively reallocates bits from layers with low $\Psi_\ell$ to those with high $\Psi_\ell$ until the bit budget is fully utilized or no further improvements can be achieved. This inner-loop allocation complements the outer-loop threshold optimization and sets the stage for the iterative routine described in Section 4.4.

## 4.4 ITERATIVE OPTIMIZATION WITH SELF-DISTILLATION RECOVERY

Finally, we integrate outer-loop threshold adaptation and inner-loop precision reallocation into an iterative routine. Each round consists of three steps: (i) updating exit thresholds $\boldsymbol{\phi}$ via grid search given a fixed bit allocation $\boldsymbol{b}$, (ii) reallocating bit-widths $\boldsymbol{b}$ based on risk–cost ratios $\Psi_\ell$, and (iii) applying a lightweight recovery step on model weights $\theta$ to mitigate accuracy degradation caused by quantization and early exiting.

In step (iii), we adopt a self-distillation scheme for accuracy recovery. Given an input $x$ with label $y$, the full-precision model $f_\theta$ produces logits $z = f_\theta(x)$, while the quantized early-exit model $f_{\theta, \boldsymbol{b}, \boldsymbol{\phi}}$ produces logits $\tilde{z} = f_{\theta, \boldsymbol{b}, \boldsymbol{\phi}}(x)$. We define the corresponding probability distributions under temperature $T$ as $p(x; f_\theta, T) = \text{softmax}\left(\frac{z}{T}\right)$ and $p(x; f_{\theta, \boldsymbol{b}, \boldsymbol{\phi}}, T) = \text{softmax}\left(\frac{\tilde{z}}{T}\right)$. The recovery loss combines ground-truth supervision with distribution alignment via KL divergence:

$$\mathcal{L}(\theta; x, y) = \mathcal{L}_{\text{CE}}(p(x; f_{\theta, \boldsymbol{b}, \boldsymbol{\phi}}, 1), y) + \beta T^2 \cdot \text{KL}\Big(p(x; f_\theta, T) \,\|\, p(x; f_{\theta, \boldsymbol{b}, \boldsymbol{\phi}}, T)\Big), \tag{12}$$

where $T > 1$ controls the softness of the probability distributions, and $\beta$ balances the two objectives. This recovery step mitigates quantization-induced distortions by aligning the predictions of the quantized model with those of the full-precision model. Together with threshold and precision updates, it ensures that exit heads adapt smoothly to noise while maintaining consistency with full-precision behavior.

## 5 THEORETICAL ANALYSIS

Here, we first formally analyze the mutual interference between quantization and early exiting. We then demonstrate that the bi-level optimization in MAQEE adapts precision to utilization by leveraging APQ, thereby outperforming static FPQ and MPQ under early exiting.

**Quantization Undermines Early Exiting.** We first show that quantization perturbs the logits and hence undermines early exiting. For each layer $\ell$, let $\Delta_\ell(x) = \rho_\ell(x) - \phi_\ell$ denote the confidence margin relative to the exit threshold $\phi_\ell$. When this margin is small, even mild quantization noise can overturn the exit decision.

**Theorem 1.** *Consider a symmetric $b_\ell$-bit affine quantizer applied to activations uniformly distributed in $[-\alpha_\ell, \alpha_\ell]$. The resulting mean-squared error is $\varepsilon_\ell(b_\ell) = \frac{\alpha_\ell^2}{3 \cdot 2^{2b_\ell}}$, which decays exponentially as the bit-width $b_\ell$ increases. Consequently, the probability of mis-exit at layer $\ell$ grows monotonically with $\varepsilon_\ell(b_\ell)$.*

When $\Delta_\ell(x) > 0$, quantization noise can erase a valid exit by lowering the confidence; when $\Delta_\ell(x) < 0$, it can trigger a false exit by artificially raising the confidence; and when $\Delta_\ell(x) = 0$, any perturbation flips the decision. As the bit-width decreases, $\varepsilon_\ell(b_\ell)$ grows, and both types of errors become more frequent. Thus, when fragile early-exit decision margins coincide with low-precision quantization, the exit mechanism becomes highly unstable, leading to the severe accuracy degradation observed in practice, as captured by our BSR in equation 7.

**Early Exiting Undermines Quantization.** While quantization noise destabilizes early exiting, the reverse also holds: input-dependent early exiting weakens the effectiveness of quantization. For a bit allocation $\boldsymbol{b}$, define the per-layer error $D_\ell(b_\ell) \geq 0$ (e.g., MSE or SNR loss) and the cumulative error $F_{\boldsymbol{b}}(k) := \sum_{\ell=1}^k D_\ell(b_\ell)$, which measures the total error after $k$ executed layers. This leads to the following theorem.

**Theorem 2.** *Let $K$ be the empirical exit depth under thresholds $\boldsymbol{\phi}$, with mean $\mu_K$ and variance $\sigma_K^2 > 0$. Let $F_{\boldsymbol{b}}(k)$ denote the cumulative quantization distortion under bit allocation $\boldsymbol{b}$. If $F_{\boldsymbol{b}}(k)$ is convex in depth $k$, then $\mathbb{E}[F_{\boldsymbol{b}}(K)] \geq F_{\boldsymbol{b}}(\mu_K) + \frac{1}{2} \underline{\delta}^{(2)} \sigma_K^2$, where $\sigma_K^2$ is the variance of exit depths and $\underline{\delta}^{(2)}$ is the minimum discrete curvature of $F_{\boldsymbol{b}}(k)$.*

The stochasticity of exit decisions introduces variance in stopping behavior, causing the cumulative distortion to deviate from its deterministic counterpart. This excess distortion grows at least linearly with $\sigma_K^2$ and becomes more pronounced in deeper networks due to positive curvature $\underline{\delta}^{(2)}$, since per-layer errors $D_\ell$ typically escalate more rapidly in later blocks as a result of attention mixing, residual stacking, and activation scaling (Zhou et al., 2018). Consequently, unstable early exits amplify quantization error and render static allocations suboptimal.

**MAQEE Achieves Better APQ.** Finally, to formally evaluate the benefit of joint optimization, we analyze the convergence and optimality of MAQEE under the APQ objective.

**Theorem 3.** *The bi-level optimization in MAQEE, i.e., iteratively updating the thresholds $\boldsymbol{\phi}$ and the bit allocation $\boldsymbol{b}$, strictly decreases the APQ loss (defined in Eq. 3) at each iteration and converges to a coordinate-wise optimum, moreover, with a fixed bit budget, no static FPQ or MPQ allocation outperforms MAQEE under early exiting.*

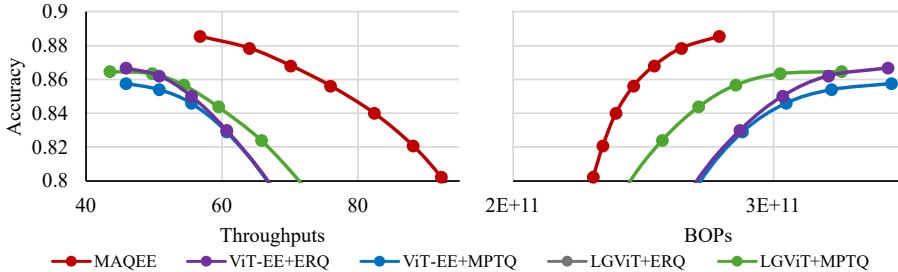

Figure 1: Accuracy–throughput/BOPs results across varying exit thresholds on CIFAR-100. The LGViT+ERQ configuration fell short of achieving 80% accuracy.

In summary, MAQEE guarantees continual improvement and convergence through bi-level optimization, and provably outperforms static FPQ and MPQ in the presence of early exiting.

## 6 EXPERIMENT

We conduct extensive experiments across multiple vision tasks and backbones to validate the effectiveness of MAQEE, comparing it with three early-exiting methods and three quantization methods on three different backbones. Overall, MAQEE achieves a 95% reduction in computation while maintaining performance comparable to the full-precision model.

### 6.1 SETUP

**Dataset and Baselines.** To evaluate MAQEE on APQ, we conduct experiments across three core vision tasks: image classification (CIFAR-100 (Krizhevsky et al., 2009), ImageNet (Deng et al., 2009)), semantic segmentation (SceneParse150 (Zhou et al., 2017)), and object detection (MS COCO (Lin et al., 2014)). We report task metrics: accuracy for classification, IoU for segmentation, and mAP for detection. We evaluate MAQEE with three backbones: DeiT (Touvron et al., 2021), ViT (Dosovitskiy et al., 2020), and Swin (Liu et al., 2021b). We compare MAQEE against early-exiting methods, including ViT-EE (Bakhtiarnia et al., 2021) and LGViT (Xu et al., 2023), as well as quantization approaches for ViTs: (i) fixed-precision quantization (FPQ), including RepQ (Li et al., 2023) and ERQ (Zhong et al., 2025); and (ii) mixed-precision quantization (MPQ), represented by MPTQ (Tai et al., 2024). All quantization is performed under the W4A4 setting (4-bit weights and activations). Here, FP4/4 enforces uniform 4-bit quantization across layers, whereas MP4/4 allows layer-wise variation in bit-widths while maintaining an average precision of 4 bits.

**Evaluation Protocol.** For a fair comparison (Shang et al., 2024), we evaluate the balance between accuracy and efficiency along two axes. First, under the *Controlled Performance* setting, we measure the exit layer $\hat{L}$ and bit operations $\hat{\text{BOPs}}$ required to reach a target accuracy, defined as quantization-only performance without EE: *87% for CIFAR-100*, *79% for ImageNet, 30% for SceneParse150*, and *55% for MS COCO*. Models that fail to meet this target are reported with $\hat{L}$ and $\hat{\text{BOPs}}$ as N/A. Second, under the *Standard Configuration* setting, baseline models use default thresholds and quantization, while MAQEE applies optimized thresholds $\phi$ and bit allocations $b$. We report the average exit layer $\bar{L}$, bit operations BOPs, and the resulting task performance. Other implementation details and hyperparameter settings are provided in Appendix C.1.

### 6.2 EXPERIMENTAL RESULTS

As shown in Table 1 and Figure 1, under both the *Controlled Performance* and *Standard Configuration* settings, MAQEE consistently achieves the best trade-off between efficiency and accuracy, reducing BOPs by over 95% while maintaining accuracy close to the full-precision model across backbones (DeiT results are given in Appendix C.2, consistent with other ViTs). In contrast, fixed-precision baselines such as RepQ and ERQ exhibit unstable behavior. RepQ, in particular, suffers from severe distribution shift, as uniformly quantizing all layers amplifies the mismatch between layer sensitivity and quantization noise (formalized in Theorem 2). As a result, RepQ frequently fails to meet the performance target and suffers substantial accuracy degradation under standard settings. Although MPTQ incorporates mixed precision and offers more flexible bit allocations, it remains restricted to a static inference path and overlooks the dynamic utility of layers under

| Bits (W/A) | Quant | EE | CIFAR-100 | | | | | ImageNet | | | | |
| | | | Controlled Perf. | | Std. Config | | | Controlled Perf. | | Std. Config | | |
| | | | $\hat{L}\downarrow$ | $\hat{B}OPs\downarrow$ | $\bar{L}\downarrow$ | $\bar{B}OPs\downarrow$ | Acc.↑ | $\hat{L}\downarrow$ | $\hat{B}OPs\downarrow$ | $\bar{L}\downarrow$ | $\bar{B}OPs\downarrow$ | Acc.↑ |
|---|---|---|---|---|---|---|---|---|---|---|---|---|
| **Swin-B** FP32 | - | - | / | / | 24 | *1.39E13* | *94.16* | / | / | 24 | *1.39E13* | *84.50* |
| | | ViT-EE | 13.59 | 9.30E12 | 16.47 | 1.09E13 | 90.80 | 16.62 | 1.23E13 | 18.23 | 1.35E13 | 83.68 |
| | | LGViT | 9.88 | 1.02E13 | 13.80 | 9.41E12 | 89.38 | 11.13 | 8.83E12 | 14.20 | 1.10E13 | 82.7 |
| FP4/4 | RepQ | ViT-EE | N/A | N/A | 18.72 | 2.12E11 | 82.23 | N/A | N/A | 20.16 | 2.27E11 | 74.75 |
| | | LGViT | N/A | N/A | 13.69 | 1.65E11 | 74.38 | N/A | N/A | 18.97 | 2.19E11 | 69.85 |
| | ERQ | ViT-EE | 23.72 | 2.69E11 | 21.26 | 2.29E11 | 83.58 | 23.46 | 2.65E11 | 20.96 | 2.31E11 | 75.45 |
| | | LGViT | N/A | N/A | 19.76 | 1.96E11 | 76.74 | N/A | N/A | 20.80 | 2.38E11 | 73.97 |
| MP4/4 | MPTQ | ViT-EE | N/A | N/A | 21.01 | 2.36E11 | 81.62 | N/A | N/A | 20.99 | 2.39E11 | 71.55 |
| | | LGViT | N/A | N/A | 16.56 | 1.90E11 | 75.07 | N/A | N/A | 21.23 | 2.42E11 | 74.35 |
| | **MAQEE** | | **12.58** | **1.53E11** | **13.79** | **1.65E11** | **89.13** | **17.60** | **2.09E11** | **18.75** | **2.12E11** | **80.78** |
| **ViT-B** FP32 | - | - | / | / | 12 | *1.80E13* | *90.47* | / | / | 12 | *1.80E13* | *82.54* |
| | | ViT-EE | 7.09 | 1.81E13 | 7.27 | 1.89E13 | 87.51 | 7.65 | 1.90E13 | 8.68 | 2.05E13 | 79.60 |
| | | LGViT | 5.90 | 1.68E13 | 6.43 | 1.76E13 | 88.51 | 6.69 | 1.80E13 | 7.08 | 1.86E13 | 80.3 |
| FP4/4 | RepQ | ViT-EE | 11.78 | 3.94E11 | 9.90 | 3.51E11 | 85.71 | N/A | N/A | 10.83 | 3.71E11 | 74.67 |
| | | LGViT | N/A | N/A | 8.08 | 3.13E11 | 73.52 | N/A | N/A | 8.74 | 3.31E11 | 72.30 |
| | ERQ | ViT-EE | 8.77 | 3.23E11 | 9.26 | 3.34E11 | 87.82 | N/A | N/A | 9.88 | 3.49E11 | 77.35 |
| | | LGViT | 11.15 | 3.81E11 | 7.68 | 3.04E11 | 77.64 | N/A | N/A | 9.12 | 3.40E11 | 73.06 |
| MP4/4 | MPTQ | ViT-EE | 8.93 | 3.27E11 | 8.71 | 3.22E11 | 85.41 | N/A | N/A | 10.26 | 3.58E11 | 76.58 |
| | | LGViT | 10.80 | 3.74E11 | 8.05 | 3.14E11 | 84.44 | 10.25 | 3.68E11 | 9.33 | 3.45E11 | 77.53 |
| | **MAQEE** | | **5.74** | **2.52E11** | **6.07** | **2.66E11** | **88.12** | **7.04** | **2.85E11** | **7.26** | **2.93E11** | **79.41** |

Table 1: Performance on Image Classification.

| Method | Segmentation (SceneParse150) | | | | | Detection (MS COCO) | | | | |
| | Controlled Perf. | | Std. Config | | | Controlled Perf. | | Std. Config | | |
| | $\hat{L}\downarrow$ | $\hat{B}OPs\downarrow$ | $\bar{L}\downarrow$ | $\bar{B}OPs\downarrow$ | IoU↑ | $\hat{L}\downarrow$ | $\hat{B}OPs\downarrow$ | $\bar{L}\downarrow$ | $\bar{B}OPs\downarrow$ | mAP@0.5↑ |
|---|---|---|---|---|---|---|---|---|---|---|
| **ViT-B** *FP32* | / | / | 12 | 1.89E13 | 34.14 | / | / | 12 | 1.95E13 | 68.23 |
| ViT-EE+ERQ | N/A | N/A | 8.10 | 3.17E+11 | 27.34 | N/A | N/A | 8.72 | 3.34E11 | 40.42 |
| ViT-EE+MPQ | 11.46 | 3.94E+11 | 8.08 | 3.16E+11 | 26.52 | N/A | N/A | 8.59 | 3.30E11 | 41.04 |
| **MAQEE** | **5.74** | **2.60E+11** | **6.79** | **2.86E+11** | **31.20** | **10.40** | **3.77E11** | **8.68** | **3.32E11** | **50.83** |

Table 2: Performance on Semantic Segmentation and Object Detection.

early-exiting. Consequently, MPTQ often selects unnecessarily deep exits or even disables early-exiting. By contrast, MAQEE jointly evaluates the decision-level risk (PGR and BSR) and the representation-level risk (SQNR$^{-1}$ and QID), thus effectively mitigating both premature and delayed exits. This leads to more stable exit behavior and up to a 50% reduction in exit depth and BOPs compared to all quantization baselines. Since ViTs are widely used for different downstream tasks, we also evaluate MAQEE on semantic segmentation and object detection in Table 2. Overall, MAQEE delivers superior results on these challenging benchmarks, improving accuracy by at least 15% under the standard configuration and providing up to $2\times$ speedup at the target performance. In detection, however, early exiting combined with quantization causes a notable accuracy drop, as detection depends heavily on mid-level, multi-scale features encoding global context. Early exits truncate these pathways, while quantization distorts regression and classification outputs, with both effects propagating downstream and leading to mislocalization and unstable predictions.

## 7 CONCLUSION

In this work, we introduced **Mutual Adaptive Quantization with Early Exiting (MAQEE)**, a bi-level optimization framework for efficient ViT inference under the **Amortized-Precision Quantization (APQ)** paradigm. Our theoretical analysis provides stability and convergence guarantees, proving that no static FPQ or MPQ scheme can match MAQEE under early exiting. Experiments on classification, segmentation, and detection demonstrate that MAQEE consistently achieves superior accuracy–efficiency trade-offs, reducing BOPs by over 95% while maintaining or even surpassing full-precision accuracy. These results highlight the importance of jointly optimizing precision and dynamic execution, pointing toward a practical path for low-latency ViT deployment in resource-constrained environments. Looking ahead, extending MAQEE to multimodal transformers and hardware-aware settings offers exciting opportunities to further advance efficient deep learning.

## REPRODUCIBILITY STATEMENT

All theoretical results, including formal proofs of convergence, optimality, and hardness (Theorems 1–4), are presented in detail in the Appendix to facilitate independent verification. For experimental evaluation, we provide a complete description of datasets (CIFAR-100, ImageNet, SceneParse150, and MS COCO) and evaluation metrics in Section 6, with additional hyperparameters and training details reported in Appendix C.1. To further improve transparency, we release our implementation as anonymous supplementary material at `https://anonymous.4open.science/r/MAQEE`. This includes training scripts, quantization routines, and code for mutual adaptive optimization. For theoretical claims, all assumptions are explicitly stated, and proofs are provided for key results regarding APQ and MAQEE.

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

## A  DETAILED PROOF

### A.1  PROOF OF THEOREM 4

**Theorem 4.** *The APQ bi-level optimization is NP-hard. Moreover, each level is NP-hard on its own: (i) with fixed exit thresholds $\phi$, the inner bit-width allocation is NP-hard; and (ii) with fixed bit-widths $b$, the outer exit-threshold optimization is also NP-hard.*

*Proof.* We first prove that the inner-level bit-width allocation is NP-hard when the thresholds $\phi$ are fixed (Lemma 1), and then show that the outer-level threshold optimization is NP-hard when the bit-widths $b$ are fixed (Lemma 2).

**Lemma 1.** *Fix any thresholds $\phi$. Consider the decision problem: "Does there exist a bit allocation $b \in \mathcal{B}$ (e.g., discrete per-layer bit choices) such that the expected complexity $\mathbb{E}_x[\sum_\ell u_\ell(x; \phi) c_\ell(x, b_\ell)]$ is at most $B$ while the expected loss $\mathbb{E}_{(x,y)}[\mathcal{L}_{CE}(f_{\theta,b,\phi}(x), y)]$ is at most $A$?" This problem is NP-hard.*

*Proof.* Reduce from the classical *0–1 Knapsack*. Given items $\{1, \ldots, L\}$ with values $v_\ell > 0$, weights $w_\ell > 0$, and capacity $W$, ask whether there exists a subset $S$ with $\sum_{\ell \in S} w_\ell \leq W$ and $\sum_{\ell \in S} v_\ell \geq V$. Construct an APQ instance with $L$ layers and a *binary* bit-choice set $\mathcal{B}_\ell = \{b_\ell^{(0)}, b_\ell^{(1)}\}$ per layer: i) Complexity mapping: set $c_\ell(\cdot)$ and (fixed) utilization $u_\ell(\cdot; \phi)$ so that $\mathbb{E}_x[u_\ell c_\ell(x, b_\ell^{(1)})] - \mathbb{E}_x[u_\ell c_\ell(x, b_\ell^{(0)})] = w_\ell$. ii) Accuracy mapping: choose a dataset and a head such that turning on the high-bit option at layer $\ell$ reduces the loss by exactly $v_\ell$ in expectation (e.g., via a separable surrogate $D_\ell(b_\ell)$ scaled appropriately). Set $B = \sum_\ell \mathbb{E}_x[u_\ell c_\ell(x, b_\ell^{(0)})] + W$ and $A = L_0 - V$, where $L_0$ is the loss under all baselines $b_\ell^{(0)}$. Then selecting $S = \{\ell : b_\ell = b_\ell^{(1)}\}$ is feasible iff the knapsack instance is feasible. Hence NP-hardness follows. □

**Lemma 2.** *Fix any bit-widths $b$. Consider the decision problem: "Does there exist thresholds $\phi$ such that the expected complexity $\mathbb{E}_x[T(\phi)]$ is at most $B$ while the expected loss $\mathbb{E}_{(x,y)}[\mathcal{L}_{CE}(f_{\theta,b,\phi}(x), y)]$ is at most $A$?" This problem is NP-hard.*

*Proof.* Reduce from *0–1 Knapsack*. Take a two-exit cascade (layers 1 and 2) with fixed logits/softmax scores precomputed on a finite dataset $\{x_i\}_{i=1}^n$. Let $C_1(x_i) \in (0, 1)$ be the confidence at the first exit and suppose the final exit (layer 2) always predicts correctly with unit cost, while exiting at layer 1 costs zero additional depth but may misclassify some points. Associate each item $i$ with a sample $x_i$ and set: i) If $x_i$ exits at layer 1 (i.e., $C_1(x_i) \geq \phi_1$), we *save* expected depth $w_i > 0$ (benefit), but we incur a loss penalty $p_i \geq 0$ if that early decision is wrong. ii) If it proceeds to layer 2 (i.e., $C_1(x_i) < \phi_1$), we pay depth $w_i$ but incur no penalty.

Choose the surrogate loss so that total early-exit penalty equals $\sum_{i \in S} p_i$, where $S = \{i : C_1(x_i) \geq \phi_1\}$. Then the constraints $\mathbb{E}_x[T(\phi)] \leq B$ and $\mathbb{E}[\mathcal{L}_{CE}] \leq A$ become $\sum_{i \in S} w_i \geq V$ and $\sum_{i \in S} p_i \leq W$, exactly the knapsack feasibility test. Thus finding $\phi_1$ is NP-hard. The argument extends to multiple exits by letting only the first threshold be active. □

Concluding Lemmas 1 and 2, each level already involves an NP-hard subproblem, even when the other level is fixed. Consequently, the joint bi-level problem is at least as hard as solving either level in isolation, thereby implying NP-hardness. Hence, the theorem follows. □

### A.2  PROOF OF THEOREM 1

**Theorem 1.** *Consider a symmetric $b_\ell$-bit affine quantizer applied to activations uniformly distributed in $[-\alpha_\ell, \alpha_\ell]$. The resulting mean-squared error is $\varepsilon_\ell(b_\ell) = \frac{\alpha_\ell^2}{3 \cdot 2^{2b_\ell}}$, which decays exponentially as the bit-width $b_\ell$ increases. Consequently, the probability of mis-exit at layer $\ell$ grows monotonically with $\varepsilon_\ell(b_\ell)$.*

*Proof.* Consider activations $x$ uniformly distributed on $[-\alpha_\ell, \alpha_\ell]$ and a symmetric $b_\ell$-bit affine quantizer on that range. Let the (per-layer) mean-squared quantization error be $\varepsilon_\ell(b_\ell)$ and let the exit rule

at layer $\ell$ compare a confidence statistic $s_\ell$ to a threshold $\tau_\ell$, with margin $r_\ell := s_\ell - \tau_\ell$. We show that the mis-exit probability

$$p_\ell := \Pr\big[\,\text{sign}(r_\ell) \neq \text{sign}(r_\ell + \delta_\ell)\big] \tag{13}$$

is monotonically increasing in $\varepsilon_\ell(b_\ell)$ and therefore monotonically decreasing in $b_\ell$.

A $b_\ell$-bit symmetric affine quantizer over $[-\alpha_\ell, \alpha_\ell]$ has step

$$\Delta_\ell \;=\; \frac{2\alpha_\ell}{2^{b_\ell}}. \tag{14}$$

Under the standard high-resolution model, the quantization error $\eta_\ell := \hat{x} - x$ is uniform on $\big[-\frac{\Delta_\ell}{2}, \frac{\Delta_\ell}{2}\big]$, independent of $x$, with

$$\text{Var}(\eta_\ell) \;=\; \frac{\Delta_\ell^2}{12} \;=\; \frac{4\alpha_\ell^2}{12 \cdot 2^{2b_\ell}} \;=\; \frac{\alpha_\ell^2}{3 \cdot 2^{2b_\ell}} \;=:\; \varepsilon_\ell(b_\ell). \tag{15}$$

Thus $\varepsilon_\ell(b_\ell)$ decays exponentially in $b_\ell$.

Let $r_\ell = s_\ell - \tau_\ell$ be the noise-free margin and let quantization perturb the statistic by an additive error $\delta_\ell$, so the noisy margin is $r_\ell + \delta_\ell$. A mis-exit occurs exactly when the sign flips:

$$\{\text{mis-exit}\} \;\Longleftrightarrow\; \text{sign}(r_\ell) \neq \text{sign}(r_\ell + \delta_\ell). \tag{16}$$

Condition on a fixed magnitude $r := |r_\ell| > 0$. By symmetry of $\delta_\ell$,

$$p_\ell(\Delta_\ell; r) \;=\; \Pr[\delta_\ell \leq -r] \;=\; F_{\delta_\ell}(-r), \tag{17}$$

where $F_{\delta_\ell}$ is the CDF of $\delta_\ell$. For smooth $s_\ell(\cdot)$, $\delta_\ell$ scales linearly with the quantizer step, so write $\delta_\ell = \Delta_\ell U$, where $U$ is a zero-mean, symmetric r.v. independent of $\Delta_\ell$ (e.g., $U \sim \text{Unif}[-\frac{1}{2}, \frac{1}{2}]$). Hence

$$p_\ell(\Delta_\ell; r) \;=\; \Pr\big[U \leq -r/\Delta_\ell\big] \;=\; F_U\left(-\frac{r}{\Delta_\ell}\right). \tag{18}$$

Since $F_U$ is increasing and $-r/\Delta_\ell$ moves toward $0$ as $\Delta_\ell$ increases, $p_\ell(\Delta_\ell; r)$ is monotonically increasing in $\Delta_\ell$. If $F_U$ has density $f_U \geq 0$, this is explicit:

$$\frac{\partial}{\partial \Delta_\ell} p_\ell(\Delta_\ell; r) \;=\; f_U\left(-\frac{r}{\Delta_\ell}\right) \frac{r}{\Delta_\ell^2} \;\geq\; 0. \tag{19}$$

Finally, since $\varepsilon_\ell = \Delta_\ell^2/12$, scaling $\Delta_\ell \mapsto c\Delta_\ell$ is equivalent to $\varepsilon_\ell \mapsto c^2 \varepsilon_\ell$, so $p_\ell$ is monotonically increasing in $\varepsilon_\ell$. Because

$$\varepsilon_\ell(b_\ell) \;=\; \frac{\alpha_\ell^2}{3 \cdot 2^{2b_\ell}} \tag{20}$$

decreases with $b_\ell$, the mis-exit probability decreases with $b_\ell$ and increases with $\varepsilon_\ell(b_\ell)$. The theorem follows. $\qquad\square$

## A.3 PROOF OF COROLLARY 1

**Corollary 1.** *Layers or samples with small margins $\Delta_\ell(x)$ are inherently unstable under low-bit quantization, since even minor perturbations can flip exit outcomes.*

*Proof.* Let $\Delta_\ell(x)$ be the noise-free exit margin at layer $\ell$ for sample $x$, and let the quantizer induce an additive perturbation $\delta_\ell(x)$ with $\mathbb{E}[\delta_\ell(x)] = 0$ and a symmetric distribution. The mis-exit event is a sign flip:

$$p_\ell(x) = \Pr\big[\,\text{sign}(\Delta_\ell(x)) \neq \text{sign}(\Delta_\ell(x) + \delta_\ell(x))\big] \;=\; \Pr\big[\delta_\ell(x) \leq -|\Delta_\ell(x)|\big] \;=\; F_{\delta_\ell}(-|\Delta_\ell(x)|), \tag{21}$$

where $F_{\delta_\ell}$ is the CDF of $\delta_\ell(x)$, hence $p_\ell(x)$ is monotonically decreasing in $|\Delta_\ell(x)|$ and monotonically increasing in the noise scale. In the uniform-error (affine quantizer) model, if the quantizer step is $\Delta_q$ and the margin statistic is locally $L_\ell$-Lipschitz so that $\delta_\ell(x) = L_\ell \eta$ with $\eta \sim \text{Unif}\big[-\frac{\Delta_q}{2}, \frac{\Delta_q}{2}\big]$, then $\delta_\ell(x) \sim \text{Unif}[-a, a]$ with $a = L_\ell \Delta_q/2$, yielding

$$p_\ell(x) \;=\; \max\left\{0, \; \frac{1}{2} - \frac{|\Delta_\ell(x)|}{L_\ell \, \Delta_q}\right\}, \tag{22}$$

so whenever $|\Delta_\ell(x)| < L_\ell \Delta_q$ the flip probability is strictly positive and decreases linearly as the margin grows. More generally, if $\delta_\ell(x)$ is sub-Gaussian with proxy variance $\sigma_\ell^2$ (e.g., $\sigma_\ell^2 = c\, L_\ell^2 \varepsilon_\ell$ with $\varepsilon_\ell$ the per-layer MSE of the quantizer), then the one-sided tail bound gives

$$p_\ell(x) \;=\; \Pr\big[\delta_\ell(x) \leq -|\Delta_\ell(x)|\big] \;\leq\; \exp\!\left(-\frac{\Delta_\ell(x)^2}{2\sigma_\ell^2}\right), \tag{23}$$

which is increasing in the noise scale $\sigma_\ell^2$ (or $\varepsilon_\ell$) and decreasing in $|\Delta_\ell(x)|$. Therefore, layers or samples with small margins $\Delta_\ell(x)$ are inherently unstable under low-bit quantization, since even minor perturbations can flip exit outcomes. The corollary follows. $\qquad\square$

## A.4 PROOF OF THEOREM 2

**Theorem 2.** *Let $K$ be the empirical exit depth under thresholds $\phi$, with mean $\mu_K$ and variance $\sigma_K^2 > 0$. Let $F_{\boldsymbol{b}}(k)$ denote the cumulative quantization distortion under bit allocation $\boldsymbol{b}$. If $F_{\boldsymbol{b}}(k)$ is convex in depth $k$, then $\mathbb{E}[F_{\boldsymbol{b}}(K)] \geq F_{\boldsymbol{b}}(\mu_K) + \frac{1}{2}\underline{\delta}^{(2)}\sigma_K^2$, where $\sigma_K^2$ is the variance of exit depths and $\underline{\delta}^{(2)}$ is the minimum discrete curvature of $F_{\boldsymbol{b}}(k)$.*

*Proof.* Let $K$ be the (integer–valued) random exit depth with mean $\mu_K := \mathbb{E}[K]$ and variance $\sigma_K^2 := \mathrm{Var}(K)$. Write the discrete forward differences of $F_{\boldsymbol{b}}$ as

$$\Delta F_{\boldsymbol{b}}(k) := F_{\boldsymbol{b}}(k) - F_{\boldsymbol{b}}(k-1) = D_k(b_k),$$

so the discrete second difference is

$$\Delta^2 F_{\boldsymbol{b}}(k) := \Delta F_{\boldsymbol{b}}(k) - \Delta F_{\boldsymbol{b}}(k-1) = D_k(b_k) - D_{k-1}(b_{k-1}).$$

By assumption, $F_{\boldsymbol{b}}$ is (discretely) convex and its discrete curvature is bounded below:

$$\Delta^2 F_{\boldsymbol{b}}(k+1) - \Delta^2 F_{\boldsymbol{b}}(k) = D_{k+1}(b_{k+1}) - 2D_k(b_k) + D_{k-1}(b_{k-1}) \;\geq\; \underline{\delta}^{(2)} \quad \text{for all } k \text{ in the support of } K.$$

This means $F_{\boldsymbol{b}}$ is $m$-strongly convex on $\mathbb{Z}$ with parameter $m := \underline{\delta}^{(2)}$ in the discrete sense: for any $x \in \mathbb{R}$ and $k \in \mathbb{Z}$ there exists a subgradient $g_x \in \partial \widetilde{F}_{\boldsymbol{b}}(x)$ of a convex extension $\widetilde{F}_{\boldsymbol{b}} : \mathbb{R} \to \mathbb{R}$ satisfying

$$F_{\boldsymbol{b}}(k) \;\geq\; \widetilde{F}_{\boldsymbol{b}}(x) + g_x\,(k-x) + \frac{m}{2}\,(k-x)^2. \tag{24}$$

(One convenient extension is the standard piecewise–linear/quadratic convex interpolation, whose second derivative in the distributional sense is bounded below by $m$; then Eq. 24 is the usual strong convexity inequality.)

Apply Eq. 24 with $x = \mu_K$ and take expectations with respect to $K$:

$$\mathbb{E}\big[F_{\boldsymbol{b}}(K)\big] \;\geq\; \widetilde{F}_{\boldsymbol{b}}(\mu_K) + g_{\mu_K}\,\mathbb{E}[K - \mu_K] + \frac{m}{2}\,\mathbb{E}\big[(K-\mu_K)^2\big]. \tag{25}$$

The middle term vanishes since $\mathbb{E}[K - \mu_K] = 0$, and $\mathbb{E}\big[(K-\mu_K)^2\big] = \sigma_K^2$. Moreover, by construction $\widetilde{F}_{\boldsymbol{b}}(\mu_K) = F_{\boldsymbol{b}}(\mu_K)$ when $\mu_K \in \mathbb{Z}$, and in general $\widetilde{F}_{\boldsymbol{b}}$ coincides with $F_{\boldsymbol{b}}$ at integer points while preserving convexity, so the bound reads

$$\mathbb{E}\big[F_{\boldsymbol{b}}(K)\big] \;\geq\; F_{\boldsymbol{b}}(\mu_K) + \frac{m}{2}\,\sigma_K^2 \;=\; F_{\boldsymbol{b}}(\mu_K) + \frac{1}{2}\,\underline{\delta}^{(2)}\,\sigma_K^2. \tag{26}$$

The theorem follows. $\qquad\square$

## A.5 PROOF OF COROLLARY 2

**Corollary 2.** *Under input-dependent early exiting, no static bit allocation $\boldsymbol{b}$ can be uniformly optimal, since exit randomness inevitably amplifies quantization error in convex regimes.*

*Proof.* Let $K$ be the random exit depth induced by thresholds $\phi$ and an input $x$, and let $F_{\boldsymbol{b}}(k)$ denote the cumulative quantization distortion up to depth $k$ under a static bit allocation $\boldsymbol{b}$. Assume the convex (discrete) regime with minimum curvature

$$\underline{\delta}^{(2)}(\boldsymbol{b}) \;:=\; \min_k \big(D_{k+1}(b_{k+1}) - 2D_k(b_k) + D_{k-1}(b_{k-1})\big) \;>\; 0. \tag{27}$$

For the distribution $\mathcal{P}$ of $K$, write $\mu_K = \mathbb{E}_{\mathcal{P}}[K]$ and $\sigma_K^2 = \mathrm{Var}_{\mathcal{P}}(K)$. By discrete strong convexity (Jensen with a curvature correction),

$$\mathbb{E}_{\mathcal{P}}\big[F_{\boldsymbol{b}}(K)\big] \geq F_{\boldsymbol{b}}(\mu_K) + \tfrac{1}{2}\underline{\delta}^{(2)}(\boldsymbol{b})\,\sigma_K^2. \tag{28}$$

Define the performance functional for $\boldsymbol{b}$ under $\mathcal{P}$ as

$$\mathcal{J}(\boldsymbol{b}\,;\mathcal{P}) := \mathbb{E}_{\mathcal{P}}\big[F_{\boldsymbol{b}}(K)\big]. \tag{29}$$

Consider two exit-depth distributions $\mathcal{P}_1, \mathcal{P}_2$ with the same mean $\mu_K$ but different variances $0 \leq v_1 < v_2$. From equation 28,

$$\mathcal{J}(\boldsymbol{b}\,;\mathcal{P}_i) \geq F_{\boldsymbol{b}}(\mu_K) + \tfrac{1}{2}\underline{\delta}^{(2)}(\boldsymbol{b})\,v_i, \qquad i \in \{1,2\}. \tag{30}$$

Optimization over $\boldsymbol{b}$ therefore faces a bias–curvature trade-off: for small-variance $\mathcal{P}_1$ the leading term is $F_{\boldsymbol{b}}(\mu_K)$, while for large-variance $\mathcal{P}_2$ the curvature penalty $\tfrac{1}{2}\underline{\delta}^{(2)}(\boldsymbol{b})\,v_2$ dominates. Let

$$\boldsymbol{b}_1 \in \arg\min_{\boldsymbol{b}} F_{\boldsymbol{b}}(\mu_K), \tag{31}$$

and

$$\boldsymbol{b}_2 \in \arg\min_{\boldsymbol{b}} \underline{\delta}^{(2)}(\boldsymbol{b}). \tag{32}$$

In any nondegenerate convex regime there exist allocations with different curvatures (changing $\boldsymbol{b}$ reshapes the per-depth distortions $D_k(b_k)$), hence typically $\boldsymbol{b}_1 \neq \boldsymbol{b}_2$. Choose $v_2$ sufficiently large so that, for all $\boldsymbol{b}$,

$$F_{\boldsymbol{b}_1}(\mu_K) - F_{\boldsymbol{b}}(\mu_K) \ll \tfrac{1}{2}\big(\underline{\delta}^{(2)}(\boldsymbol{b}) - \underline{\delta}^{(2)}(\boldsymbol{b}_2)\big)\,v_2, \tag{33}$$

which forces $\boldsymbol{b}_2$ to be optimal for $\mathcal{P}_2$, while $\boldsymbol{b}_1$ is optimal for $\mathcal{P}_1$. Consequently, there is no single static $\boldsymbol{b}$ that minimizes $\mathcal{J}(\boldsymbol{b}\,;\mathcal{P})$ for both $\mathcal{P}_1$ and $\mathcal{P}_2$. Since input-dependent early exiting induces input-dependent exit-depth distributions $\mathcal{P}_x$ with generally different variances, any fixed $\boldsymbol{b}$ cannot be uniformly optimal across inputs. Intuitively, exit randomness amplifies quantization error via the convex penalty $\tfrac{1}{2}\underline{\delta}^{(2)}(\boldsymbol{b})\sigma_K^2$ in equation 28; because $\underline{\delta}^{(2)}(\boldsymbol{b})$ itself depends on $\boldsymbol{b}$, the optimal trade-off necessarily varies with the variance of $K$, precluding a uniform optimum. The corollary follows. $\qquad\square$

## A.6 PROOF OF THEOREM 3

**Theorem 3.** *The bi-level optimization in MAQEE, i.e., iteratively updating the thresholds $\phi$ and the bit allocation $\boldsymbol{b}$, strictly decreases the APQ loss (defined in Eq. 3) at each iteration and converges to a coordinate-wise optimum, moreover, with a fixed bit budget, no static FPQ or MPQ allocation outperforms MAQEE under early exiting.*

*Proof.* Let the network have $L$ layers with early-exit heads at layers $1, \ldots, L-1$. Let $\phi = (\phi_1, \ldots, \phi_{L-1})$ be the exit thresholds, and let $s_\ell(x)$ be the confidence statistic at exit $\ell$ for input $x$ drawn from the data distribution. Define the random exit depth

$$K_{\phi}(x) := \min\{\ell \in \{1, \ldots, L\} : s_\ell(x) \geq \phi_\ell\}, \quad K_{\phi}(x) = L \text{ if none triggers.} \tag{34}$$

Write $K_{\phi}$ when the dependence on $x$ is implicit. Set $\mu_{K_{\phi}} := \mathbb{E}[K_{\phi}]$, $\sigma_{K_{\phi}}^2 := \mathrm{Var}(K_{\phi})$, and $T(\phi) := \mathbb{E}[K_{\phi}]$.

Let the bit allocation be $\boldsymbol{b} = (b_1, \ldots, b_L)$ with budget $\sum_{\ell=1}^{L} b_\ell = B$. For each layer $\ell$, let $D_\ell(b_\ell) \geq 0$ denote the expected increase of the task loss from quantizing layer $\ell$ at $b_\ell$ bits (holding shallower layers fixed). Assume $D_\ell(\cdot)$ is nonincreasing and convex. Define the cumulative distortion up to depth $k$,

$$F_{\boldsymbol{b}}(k) := \sum_{\ell=1}^{k} D_\ell(b_\ell), \tag{35}$$

and its discrete curvature lower bound

$$\delta^{(2)}(\boldsymbol{b}) := \min_{2 \leq k \leq L-1} \big(D_{k+1}(b_{k+1}) - 2D_k(b_k) + D_{k-1}(b_{k-1})\big) > 0. \tag{36}$$

The APQ loss is

$$\mathcal{L}(\boldsymbol{b}, \boldsymbol{\phi}) := \mathbb{E}[F_{\boldsymbol{b}}(K_\phi)] + \lambda \, T(\boldsymbol{\phi}). \tag{37}$$

Assume the discrete strong-convexity variance bound holds:

$$\mathbb{E}[F_{\boldsymbol{b}}(K_\phi)] \geq F_{\boldsymbol{b}}(\mu_{K_\phi}) + \tfrac{1}{2} \, \delta^{(2)}(\boldsymbol{b}) \, \sigma^2_{K_\phi}. \tag{38}$$

Assume that a small change $\phi \to \phi'$ that moves thresholds away from near-boundary regions strictly reduces $\sigma^2_{K_\phi}$ while changing $T(\boldsymbol{\phi})$ by $o(1)$. Assume also that, under the fixed budget, moving one bit from layer $i$ to $j$ according to a risk–cost rule strictly decreases $F_{\boldsymbol{b}}(\mu_{K_\phi})$ and does not increase $\delta^{(2)}(\boldsymbol{b})$. With fixed $\boldsymbol{b}$, using the variance bound,

$$\mathcal{L}(\boldsymbol{b}, \boldsymbol{\phi}) \geq F_{\boldsymbol{b}}(\mu_{K_\phi}) + \tfrac{1}{2} \, \delta^{(2)}(\boldsymbol{b}) \, \sigma^2_{K_\phi} + \lambda \, T(\boldsymbol{\phi}). \tag{39}$$

Choose a small update $\phi \to \phi'$ that reduces $\sigma^2_{K_\phi}$ and changes $T(\boldsymbol{\phi})$ by $o(1)$. Then

$$\mathcal{L}(\boldsymbol{b}, \boldsymbol{\phi}') - \mathcal{L}(\boldsymbol{b}, \boldsymbol{\phi}) \lesssim \tfrac{1}{2} \, \delta^{(2)}(\boldsymbol{b}) \big( \sigma^2_{K_{\phi'}} - \sigma^2_{K_\phi} \big) + \lambda \, o(1) \; < \; 0. \tag{40}$$

With fixed $\phi'$, reallocate one bit from layer $i$ to $j$ under the budget. With $T(\phi')$ unchanged and $\Delta\delta^{(2)}(\boldsymbol{b}) \leq 0$, the first-order change satisfies

$$\Delta\mathcal{L} \approx \partial_{b_j} F_{\boldsymbol{b}}(\mu_{K_{\phi'}}) - \partial_{b_i} F_{\boldsymbol{b}}(\mu_{K_{\phi'}}) + \tfrac{1}{2} \, \sigma^2_{K_{\phi'}} \, \Delta\delta^{(2)}(\boldsymbol{b}) \; < \; 0. \tag{41}$$

Each iteration (threshold update followed by bit reallocation) strictly decreases $\mathcal{L}(\boldsymbol{b}, \boldsymbol{\phi})$. Since the feasible set $\{\boldsymbol{b} : \sum_\ell b_\ell = B, \, b_\ell \geq 0\} \times \Phi$ is compact in practice and $\mathcal{L}$ is bounded below, the sequence is monotonically decreasing and converges to a coordinate-wise optimum.

For the static suboptimality claim, choose $\phi_1, \phi_2$ with the same $\mu_{K_\phi}$ and with $0 \leq \sigma^2_{K_{\phi_1}} < \sigma^2_{K_{\phi_2}}$. For any static $\boldsymbol{b}$,

$$\mathcal{L}(\boldsymbol{b}, \boldsymbol{\phi}_2) - \mathcal{L}(\boldsymbol{b}, \boldsymbol{\phi}_1) \geq \tfrac{1}{2} \, \delta^{(2)}(\boldsymbol{b}) \big( \sigma^2_{K_{\phi_2}} - \sigma^2_{K_{\phi_1}} \big) \; > \; 0, \tag{42}$$

and because $\delta^{(2)}(\boldsymbol{b})$ depends on $\boldsymbol{b}$, the allocation favored under $\phi_1$ generally differs from that under $\phi_2$. Hence no static FPQ or MPQ allocation outperforms MAQEE under early exiting. □

## B    NOTATION TABLE

| Symbol | Description |
|---|---|
| $L$ | Number of transformer layers. |
| $B$ | Set of admissible bit-widths (e.g., $\{3, 4, 5\}$). |
| $b_\ell$ | Bit-width assigned to layer $\ell$. |
| $\mathbf{b}$ | Vector of layer bit-widths. |
| $\varphi_\ell$ | Confidence threshold at exit $\ell$. |
| $\boldsymbol{\varphi}$ | Vector of exit thresholds. |
| $z_\ell(x)$ | Logits from the exit head at layer $\ell$ for input $x$. |
| $\rho_\ell(x)$ | Exit confidence at layer $\ell$. |
| $u_\ell(x, \boldsymbol{\varphi})$ | Indicator that input $x$ reaches layer $\ell$ under $\boldsymbol{\varphi}$. |
| $u_\ell(\boldsymbol{\varphi})$ | Expected utilization of layer $\ell$. |
| $T(\boldsymbol{\varphi})$ | Expected depth (average number of executed layers). |
| $c_\ell(x, b_\ell)$ | Compute cost of layer $\ell$ at bit-width $b_\ell$ (e.g., BOPs). |
| $\text{Complexity}_{\text{APQ}}(\mathbf{b}, \boldsymbol{\varphi})$ | Amortized compute under policy $\boldsymbol{\varphi}$ and bit allocation $\mathbf{b}$. |
| $\lambda, \lambda_{\text{outer}}, \lambda_{\text{inner}}$ | Trade-off coefficients in optimization objectives. |
| $h_\ell, \tilde{h}_\ell$ | Full-precision and quantized activations at layer $\ell$. |
| $\text{SQNR}_\ell^{-1}$ | Inverse signal-to-quantization-noise ratio at layer $\ell$. |
| $\text{QID}_\ell$ | Quantization-induced drift (range distortion). |
| $\text{PGR}_\ell(\boldsymbol{\varphi})$ | Performance gap risk at exit $\ell$. |
| $\text{BSR}_\ell(\boldsymbol{\varphi})$ | Boundary sensitivity risk near threshold $\varphi_\ell$. |
| $R_\ell$ | Per-layer risk combining decision- and representation-level terms. |
| $\Psi_\ell$ | Risk–cost ratio for prioritizing precision allocation. |
| $x, y$ | Input sample and label. |
| $f_{\theta, \mathbf{b}, \boldsymbol{\varphi}}$ | Quantized network with weights $\theta$, bit allocation $\mathbf{b}$, and policy $\boldsymbol{\varphi}$. |
| $\mathcal{L}_{\text{CE}}$ | Cross-entropy loss. |
| $\text{KL}(\cdot\|\cdot), T, \beta$ | Terms in the self-distillation loss. |
| $\alpha$ | Trade-off between decision- and representation-level risks. |
| $\tau_\ell$ | Tolerance margin used in BSR. |
| $\alpha_\ell$ | Activation range parameter (used in Theorem 1). |
| $\varepsilon_\ell(b_\ell)$ | Quantization MSE for a $b_\ell$-bit uniform affine quantizer. |
| $K, \mu_K, \sigma_K^2$ | Random exit depth and its mean/variance. |
| $D_\ell(b_\ell), F_b(k)$ | Per-layer and cumulative quantization distortions. |

Table 3: Notation used throughout the paper.

## C    EXPERIMENTS

### C.1    IMPLEMENTATION DETAILS

All training are conducted on NVIDIA H100 GPUs. We adopt the SegFormer (Xie et al., 2021) head on a ViT-B for semantic segmentation, and conduct object detection experiments on YOLOS-Base (Fang et al., 2021), which is based on DeiT-B. For each model, we configure four exit heads: the first two are convolutional, and the latter two are attention-based. In the 12-layer DeiT-B and ViT-B models, the exit heads are inserted at the 4[th], 6[th], 8[th], and 10[th] layers. For the Swin model, they are placed at the 2[nd], 4[th], 14[th], and 20[th] layers, approximately uniformly distributed with respect to computational cost. All models are trained with AdamW and a cosine-decay schedule with a $1 \times 10^{-5}$ peak LR. For MPQ optimization, the process is initialized using the FPQ configuration of the layer immediately preceding the target precision. The optimization is then progressively refined toward the desired precision until convergence is achieved. We set the hyperparameters as follows: $\lambda_{\text{outer}} = 0.5, \lambda_{\text{inner}} = 0.5, \alpha = 0.75, \beta = 0.5, \tau = 0.05, \epsilon = 10^{-6}, T = 2, b_{\min} = 3$, and $b_{\max} = 5$.

Table 4: Performance on Image Classification with DeiT dataset.

| | Bits (W/A) | Quant | EE | CIFAR-100 | | | | | ImageNet | | | | |
|---|---|---|---|---|---|---|---|---|---|---|---|---|---|
| | | | | Fix Acc. | | Default $\phi$ | | | Fix Acc. | | Default $\phi$ | | |
| | | | | $\widehat{L}\downarrow$ | $\widehat{\text{BOPs}}\downarrow$ | $\bar{L}\downarrow$ | $\bar{\text{BOPs}}\downarrow$ | Acc.↑ | $\widehat{L}\downarrow$ | $\widehat{\text{BOPs}}\downarrow$ | $\bar{L}\downarrow$ | $\bar{\text{BOPs}}\downarrow$ | Acc.↑ |
| DeiT | FP32 | - | - | / | / | *12* | *1.80E13* | *91.86* | / | / | *12* | *1.80E13* | *83.10* |
| | | | ViT-EE | 7.20 | 1.84E13 | 8.10 | 1.97E13 | 89.24 | 9.59 | 2.19E13 | 8.15 | 1.97E13 | 80.51 |
| | | | LGViT | 5.47 | 1.61E13 | 6.33 | 1.70E13 | 89.03 | 6.19 | 1.72E13 | 7.18 | 1.87E13 | 81.70 |
| | FP4/4 | RepQ | ViT-EE | N/A | N/A | 9.91 | 3.59E11 | 84.79 | N/A | N/A | 11.42 | 3.85E11 | 70.58 |
| | | | LGViT | N/A | N/A | 9.37 | 3.42E11 | 75.43 | N/A | N/A | 10.03 | 3.56E11 | 69.31 |
| | | ERQ | ViT-EE | 8.77 | 3.23E11 | 9.28 | 3.35E11 | 87.86 | 11.79 | 3.91E11 | 10.86 | 3.72E11 | 77.59 |
| | | | LGViT | N/A | N/A | 9.00 | 3.34E11 | 78.51 | N/A | N/A | 9.09 | 3.37E11 | 71.32 |
| | MP4/4 | MPQ | ViT-EE | 8.81 | 3.25E11 | 8.51 | 3.17E11 | 84.57 | 11.80 | 3.94E11 | 10.56 | 3.65E11 | 76.15 |
| | | | LGViT | 11.35 | 3.93E11 | 7.76 | 3.07E11 | 83.15 | 12 | 4.09E11 | 8.86 | 3.34E11 | 73.58 |
| | | **MAQEE** | | **5.86** | **2.61E11** | **5.98** | **2.64E11** | **87.86** | **7.92** | **3.10E11** | **7.59** | **3.01E11** | **78.57** |

### C.2 DeiT Results

As shown in Table 4, DeiT delivers performance broadly comparable to ViT, consistent with their structural similarity. MAQEE preserves FP32-level exiting accuracy and enables, on average, 3–4 earlier exits at the same accuracy. Although MPQ alleviates part of the accuracy degradation of FPQ under Early Exiting, its assumption of a fixed inference path constrains exit efficiency, leaving it less effective than MAQEE.

## D Bi-Level Optimization Algorithm

**Algorithm 1** Bi-level optimization

**Require:** Parameters $\theta$; datasets $\mathcal{D}_{\text{train}}, \mathcal{D}_{\text{cal}}$; initial bit allocation $\mathbf{b}^{(0)}$; threshold grid $\Lambda$; hyper-parameters $\alpha, \beta$; accuracy drop tolerance $\epsilon_{\text{acc}}$; per-layer bit bounds $b_{\min}, b_{\max}$
**Ensure:** Thresholds $\boldsymbol{\varphi}^\star$, bit-widths $\mathbf{b}^\star$
  **Initialization:** set thresholds $\boldsymbol{\varphi}^{(0)}$ (e.g., high-confidence defaults); quantize model with $\mathbf{b}^{(0)}$; estimate utilization $u_\ell(\boldsymbol{\varphi}^{(0)})$ and expected depth $T(\boldsymbol{\varphi}^{(0)}) = \sum_\ell u_\ell(\boldsymbol{\varphi}^{(0)})$ on $\mathcal{D}_{\text{cal}}$.
  Set $\boldsymbol{\varphi} \leftarrow \boldsymbol{\varphi}^{(0)}$, $\mathbf{b} \leftarrow \mathbf{b}^{(0)}$.
  **while true do**
    $\mathbf{b}_{\text{prev}} \leftarrow \mathbf{b}$
    **Outer loop (threshold search):**
    Given $\boldsymbol{b}$:
    **for** each exit $\ell$ and candidate $\lambda \in \Lambda$ **do**
      Compute $\mathrm{PGR}_\ell(\lambda)$(equation 6) and $\mathrm{BSR}_\ell(\lambda)$(equation 7).
    **end for**
    Update thresholds $\boldsymbol{\varphi}$ by coordinate/grid search (equation 8).
    Re-estimate $u_\ell(\boldsymbol{\varphi})$ on $\mathcal{D}_{\text{cal}}$.
    **Inner loop (bit re-allocation):**
    **for** each layer $\ell$ **do**
      Compute per-layer risk $R_\ell$ (equation 10)
      Compute normalized score $\Psi_\ell$ (equation 11) by $R_\ell$ and updated $u_\ell(\boldsymbol{\varphi})$.
    **end for**
    Define feasible sets: $\mathcal{L}_\downarrow = \{\ell \mid b_\ell > b_{\min}\}, \quad \mathcal{L}_\uparrow = \{\ell \mid b_\ell < b_{\max}\}$.
    Choose release target $\ell^\downarrow = \arg\min_{\ell \in \mathcal{L}_\downarrow} \Psi_\ell$, allocation target $\ell^\uparrow = \arg\max_{\ell \in \mathcal{L}_\uparrow} \Psi_\ell$.
    Update bits (budget-conserving): $b_{\ell^\downarrow} \leftarrow b_{\ell^\downarrow} - 1; \quad b_{\ell^\uparrow} \leftarrow b_{\ell^\uparrow} + 1$.
    **Self-distillation recovery:** train on $\mathcal{D}_{\text{train}}$ with loss $\mathcal{L}(\theta; x, y)$ (equation 12).
    Evaluate $\Delta_{\text{acc}}$ on $\mathcal{D}_{\text{cal}}$.
    **Convergence check:**
    **if** $\mathbf{b} = \mathbf{b}_{\text{prev}}$ **then**
      **break**
    **end if**
  **end while**
  **return** $\boldsymbol{\varphi}^\star = \boldsymbol{\varphi}, \quad \mathbf{b}^\star = \mathbf{b}$

---

