# OpenReview forum: "Amortized-Precision Quantization for Early Exiting in Vision Transformers"
_ICLR.cc/2026/Conference — ICLR 2026 Conference Withdrawn Submission_

### Official Review · Reviewer_56KL · 2025-10-30

**Soundness:** 4
**Presentation:** 3
**Contribution:** 3
**Rating:** 4
**Confidence:** 3

**Summary:**

Vision Transformers (ViTs) achieve strong performance across various computer vision tasks, yet their large parameter count and high computational cost make deployment on resource-constrained edge devices difficult.
This paper proposes Mutual Adaptive Quantization with Early Exiting (MAQEE) — a compatibility-oriented framework that effectively integrates Quantization and Early Exiting (EE) to address this limitation.
MAQEE resolves the trade-offs between depth–precision and shallow–deep that arise when combining quantization with dynamic inference. It adaptively modifies static inference paths based on input complexity, reducing computation while maintaining accuracy.
Through bi-level optimization, the framework searches for robust exit thresholds against quantization noise and dynamically reallocates bit-widths based on layer utilization and accumulated quantization distortion.
The proposed method achieves up to 95% reduction in BOPs with minimal accuracy loss, supported by four theoretical analyses (Theorems 1–4).
Overall, this work introduces a new paradigm for efficient and robust ViT deployment under dynamic precision conditions.

**Strengths:**

- Clarified Motivation:
MAQEE explicitly identifies and formulates the mutual interference challenge between quantization and early exiting for ViT compression.
Theoretical analyses (Theorems 1 and 2) mathematically formalize this issue, and the introduction of the Amortized-Precision Quantization (APQ) paradigm — which allocates precision based on layer utilization — is conceptually novel and potentially impactful for future research.

 - Robustness of the Framework:
The proposed bi-level optimization framework systematically separates the outer loop (exit-threshold optimization) and inner loop (bit allocation), presenting a clear and logically sound design.
The introduction of specific risk metrics — Boundary Sensitivity Risk (BSR) and Quantization-Induced Drift (QID) — provides a concrete and innovative approach to stabilizing dynamic quantization, effectively addressing the stated problem.

 - Experimental Superiority:
The experiments demonstrate that PTQ-based baselines (RepQ, ERQ, MPTQ) combined with EE often fail to reach the target accuracy (reported as N/A), while MAQEE achieves the lowest average exit depth (L) and BOPs across all tasks, maintaining the highest accuracy among all methods.

**Weaknesses:**

- Lack of Comparison with QAT-based Methods:
MAQEE includes a self-distillation recovery stage requiring ground-truth labels 𝑦  which introduces additional training cost compared to PTQ. Therefore, comparisons should also include Quantization-Aware Training (QAT) methods or QAT+EE approaches such as QuEE [1], ideally reported in the main result tables (Table 1 and Table 2).
[1] Predicting Probabilities of Error to Combine Quantization and Early Exiting (QuEE), arXiv, 2024.

 - Limited Precision and Model Diversity:
The experiments mainly focus on 4-bit quantization (W4A4). Evaluations under lower (2-bit) or higher (6-bit / 8-bit) precision settings would provide a more complete understanding.
Similarly, additional results on smaller models (e.g., ViT-S, ViT-Tiny) and lightweight variants (e.g., DeiT, MobileViT) would enhance the generality of the claims.

 - Restricted Bit Range in MPQ:
In the baseline MPQ setup, the minimum and maximum bit widths are fixed at 3 and 5.
Testing with a wider bit range would further highlight MAQEE’s advantage under highly dynamic inference paths.

 - Clarity of the Overall Framework:
Although the mathematical formulation is solid, the overall pipeline of MAQEE is quite complex.
Adding a high-level figure that visualizes the entire processing flow (outer and inner loop interactions) would help non-expert readers better understand the method.

**Questions:**

The optimization superiority is mainly measured by theoretical metrics such as L (average exit layer) and BOPs.
Is there a plan to evaluate MAQEE under real hardware constraints (e.g., GPU/NPU bandwidth and cache structure) that may affect actual latency and throughput?

Considering that modern GPUs support specific precision modes (e.g., INT8, INT4), how does MAQEE perform in practical settings such as 8/4 or 8/8 quantization?
Is there any noticeable difference in real inference speed compared to static quantization?

Could the authors provide an analysis of training cost?
Since ViT-Base contains a large number of parameters, it would be valuable to report the additional computational overhead introduced by MAQEE’s bi-level optimization and self-distillation stages.

---

### Official Review · Reviewer_PBHV · 2025-10-30

**Soundness:** 2
**Presentation:** 2
**Contribution:** 2
**Rating:** 4
**Confidence:** 3

**Summary:**

This work establishes a novel research direction by integrating model quantization with early exiting, proposing the Amortized-Precision Quantization paradigm, a dynamic inference-path quantization  framework. The authors introduce Mutual Adaptive Quantization with Early Exiting to optimize two critical trade-offs between quantization and early exiting at both global and local levels. Comprehensive theoretical analysis demonstrates the effectiveness and rationality of the proposed optimization. Applied to ViTs, this approach achieves an balance between accuracy and efficiency through coordinated quantization and early exiting.

**Strengths:**

1. This paper formalizes the accuracy-efficiency trade-off inherent in integrating quantization with early exiting for Vision Transformers (ViTs), and proposes the Amortized-Precision Quantization (APQ) paradigm to shown its challenge.
2. This paper shown the effectiveness and stability of the MAQEE optimization framework.
3. Experimental results show that the method achieves superior accuracy compared to naive combinations of quantization and early exiting.

**Weaknesses:**

1. The method introduces an excessive number of hyperparameters for optimization. Although theoretical guarantees are provided for MAQEE's stability, the proliferation of hyperparameters leads to sensitivity challenges in practical implementation and limits extensibility to other models or tasks. Furthermore, the absence of comprehensive ablation studies on these hyperparameters raises concerns that the reported performance may heavily depend on extensive parameter tuning.
2. The computational cost of MAQEE's optimization process is not adequately addressed in the paper. This represents a significant consideration, as prolonged training requirements would substantially diminish the method's practical utility.
3. The paper lacks individual ablation studies analyzing the contributions of MAQEE's constituent components. Without such analysis, the effectiveness of each optimization element remains questionable.

**Questions:**

1. While the proposed method demonstrates effectiveness on the other tasks, its generalizability to other model architectures or vision tasks requires further investigation. What are the potential challenges in extending MAQEE to different domains or applications?
2. Could the authors clarify the optimization procedure of MAQEE. Specifically, whether the components are optimized jointly or sequentially? Furthermore, is there any observed interference or conflict between the optimization objectives of different components?

---

### Official Review · Reviewer_YmZW · 2025-10-31

**Soundness:** 2
**Presentation:** 2
**Contribution:** 2
**Rating:** 2
**Confidence:** 4

**Summary:**

This paper proposes APQ, a quantization paradigm that aligns bit-width allocation with layer utilization in early exiting. To implement APQ, the authors design MAQEE, a bi-level optimization framework that jointly optimizes exit thresholds via a Boundary Sensitivity Risk metric, and per-layer bit-widths via Quantization-Induced Drift. Theoretical analyses demonstrate the mutual interference between quantization and early exits and prove MAQEE’s convergence. Experiments on multiple models and benchmarks show efficiency improvement while maintaining or exceeding baseline accuracy.

**Strengths:**

* The method is well motivated and good framed.
* The paper is well-structured and clear.

**Weaknesses:**

* Experiment section needs to be significantly improved, for example, lacks of many important ablations including new introduced hyperparameters trade-off $\alpha$ and $\beta$, is the dataset selection sensitive, or is the initial bit distribution sensitive, how about the threshold grid? As the proposed method is somehow a little bit tricky, it’s very important to do comprehensive ablation study to further analyze the effectiveness.
* Lack of the default setting of hyperparameters, learning rate, training epochs and more.
* From my understanding, the process of quantization is extremely high, as it may need multiple iterations, and in each iteration, you need do a grid search (with a inner loop) and a fine-tuning step (distillation). Beside the better efficiency-accuracy trade-off after quantization, its important to provide more resources utilization details for quantization process, for an efficiency-related paper, these details are critical.
* The results are reported without variance, its better to run multiple times to verify the effect of randomness.

**Questions:**

Please refer to the weakness.

---

### Note · Authors · 2025-11-12

**Comment:**

We sincerely thank all reviewers for their constructive feedback.
To clarify, all hyperparameters in MAQEE are fixed and consistently applied across all models and tasks, with detailed configurations provided in the appendix. The quantization calibration is conducted only on a small calibration subset, followed by lightweight recovery training on a portion of the training data, ensuring that the overall optimization cost remains manageable. In the next version, we plan to extend the work with more comprehensive ablation studies, detailed analyses of training overhead, and experiments under varied quantization settings.
Given the current review timeline and the scope of necessary revisions, we have decided to withdraw the paper at this stage.

**Withdrawal Confirmation:**

I have read and agree with the venue's withdrawal policy on behalf of myself and my co-authors.

---

### Note · Authors · 2025-11-12

**Comment:**

We sincerely thank all reviewers for their constructive feedback.

To clarify, all hyperparameters in MAQEE are fixed and consistently applied across all models and tasks, with detailed configurations provided in the appendix. The quantization calibration is conducted only on a small calibration subset, followed by lightweight recovery training on a portion of the training data, ensuring that the overall optimization cost remains manageable. In the next version, we plan to extend the work with more comprehensive ablation studies, detailed analyses of training overhead, and experiments under varied quantization settings.

Given the current review timeline and the scope of necessary revisions, we have decided to withdraw the paper at this stage.

**Withdrawal Confirmation:**

I have read and agree with the venue's withdrawal policy on behalf of myself and my co-authors.